# Effectiveness of Group Problem Management Plus, a brief psychological intervention for adults affected by humanitarian disasters in Nepal: A cluster randomized controlled trial

Mark J. D. Jordans[1,2‡], Brandon A. Kohrt[2,3‡]*, Manaswi Sangraula[2], Elizabeth L. Turner[4], Xueqi Wang[4], Pragya Shrestha[2], Renasha Ghimire[2], Edith van 't Hof[5], Richard A. Bryant[6], Katie S. Dawson[6], Kedar Marahatta[7], Nagendra P. Luitel[2‡], Mark van Ommeren[5‡]

1 Center for Global Mental Health, Institute of Psychiatry, Psychology and Neuroscience, King's College London, London, United Kingdom, 2 Transcultural Psychosocial Organization (TPO) Nepal, Kathmandu, Nepal, 3 Division of Global Mental Health, Department of Psychiatry & Behavioral Sciences, The George Washington University, Washington, DC, United States of America, 4 Department of Biostatistics & Bioinformatics and Duke Global Health Institute, Duke University, Durham, North Carolina, United States of America, 5 World Health Organization, Geneva, Switzerland, 6 University of New South Wales, Sydney, Australia, 7 World Health Organization Country Office for Nepal, Kathmandu, Nepal

‡ MJDJ and BAK are joint first authors on this work. NPL and MvO are joint senior authors on this work.
* bkohrt@gwu.edu

## Abstract

### Background

Globally, 235 million people are impacted by humanitarian emergencies worldwide, presenting increased risk of experiencing a mental disorder. Our objective was to test the effectiveness of a brief group psychological treatment delivered by trained facilitators without prior professional mental health training in a disaster-prone setting.

### Methods and findings

We conducted a cluster randomized controlled trial (cRCT) from November 25, 2018 through September 30, 2019. Participants in both arms were assessed at baseline, midline (7 weeks post-baseline, which was approximately 1 week after treatment in the experimental arm), and endline (20 weeks post-baseline, which was approximately 3 months posttreatment). The intervention was Group Problem Management Plus (PM+), a psychological treatment of 5 weekly sessions, which was compared with enhanced usual care (EUC) consisting of a family psychoeducation meeting with a referral option to primary care providers trained in mental healthcare. The setting was 72 wards (geographic unit of clustering) in eastern Nepal, with 1 PM+ group per ward in the treatment arm. Wards were eligible if they were in disaster-prone regions and residents spoke Nepali. Wards were assigned to study arms based on covariate constrained randomization. Eligible participants were adult women and men 18 years of age and older who met screening criteria for psychological distress and functional impairment. Outcomes were measured at the participant level, with assessors

**Data Availability Statement:** Data are available through Duke University Research Data Repository https://doi.org/10.7924/r4gh9jq3g.

**Funding:** USAID OFDA funded this study in a grant to the World Health Organization (PI: MvO). BAK is supported by the US National Institute of Mental Health (K01MH104310, R01MH120649). The funders played no role in the design and conduct of the study; collection, management, analysis, and interpretation of the data; preparation, review, or approval of the manuscript; and decision to submit the manuscript for publication.

**Competing interests:** The authors have declared that no competing interests exist.

**Abbreviations:** AUDIT, Alcohol Use Disorders Identification Test; CIDT, Community Informant Detection Tool; CONSORT, Consolidated Standards of Reporting Trials; COVID-19, Coronavirus Disease 2019; cRCT, cluster randomized controlled trial; ENACT, ENhancing Assessment of Common Therapeutic factors; EQUIP, Ensuring Quality in Psychological Support; EUC, enhanced usual care; GHQ-12, General Health Questionnaire; ICC, intracluster correlation coefficient; mhCACI, mental health Cultural Adaptation and Contextualization for Implementation; mhGAP-IG, mental health Gap Action Programme-Intervention Guide; MSPSS, Multi-dimensional Scale of Perceived Social Support; PCL, PTSD CheckList; PHQ-9, Patient Health Questionnaire; PM+, Problem Management Plus; PTSD, posttraumatic stress disorder; RTC, Reducing Tension Checklist; SMD, standardized mean difference; SSS-8, Somatic Symptom Scale 8; TPO, Transcultural Psychosocial Organization; WHO, World Health Organization; WHODAS-II, World Health Organization Disability Assessment Schedule II.

blinded to group assignment. The primary outcome was psychological distress assessed with the General Health Questionnaire (GHQ-12). Secondary outcomes included depression symptoms, posttraumatic stress disorder (PTSD) symptoms, "heart–mind" problems, social support, somatic symptoms, and functional impairment. The hypothesized mediator was skill use aligned with the treatment's mechanisms of action. A total of 324 participants were enrolled in the control arm (36 wards) and 319 in the Group PM+ arm (36 wards). The overall sample ($N$ = 611) had a median age of 45 years (range 18–91 years), 82% of participants were female, 50% had recently experienced a natural disaster, and 31% had a chronic physical illness. Endline assessments were completed by 302 participants in the control arm (36 wards) and 303 participants in the Group PM+ arm (36 wards). At the midline assessment (immediately after Group PM+ in the experimental arm), mean GHQ-12 total score was 2.7 units lower in Group PM+ compared to control (95% CI: 1.7, 3.7, $p$ < 0.001), with standardized mean difference (SMD) of −0.4 (95% CI: −0.5, −0.2). At 3 months posttreatment (primary endpoint), mean GHQ-12 total score was 1.4 units lower in Group PM+ compared to control (95% CI: 0.3, 2.5, $p$ = 0.014), with SMD of −0.2 (95% CI: −0.4, 0.0). Among the secondary outcomes, Group PM+ was associated with endline with a larger proportion attaining more than 50% reduction in depression symptoms (29.9% of Group PM+ arm versus 17.3% of control arm, risk ratio = 1.7, 95% CI: 1.2, 2.4, $p$ = 0.002). Fewer participants in the Group PM+ arm continued to have "heart–mind" problems at endline (58.8%) compared to the control arm (69.4%), risk ratio = 0.8 (95% CI, 0.7, 1.0, $p$ = 0.042). Group PM+ was not associated with lower PTSD symptoms or functional impairment. Use of psychosocial skills at midline was estimated to explain 31% of the PM+ effect on endline GHQ-12 scores. Adverse events in the control arm included 1 suicide death and 1 reportable incidence of domestic violence; in the Group PM+ arm, there was 1 death due to physical illness. Study limitations include lack of power to evaluate gender-specific effects, lack of long-term outcomes (e.g., 12 months posttreatment), and lack of cost-effectiveness information.

## Conclusions

In this study, we found that a 5-session group psychological treatment delivered by nonspecialists modestly reduced psychological distress and depression symptoms in a setting prone to humanitarian emergencies. Benefits were partly explained by the degree of psychosocial skill use in daily life. To improve the treatment benefit, future implementation should focus on approaches to enhance skill use by PM+ participants.

## Trial registration

ClinicalTrials.gov NCT03747055.

## Author summary

### Why was this study done?

- Millions of people worldwide are affected by humanitarian emergencies such as war, environmental disasters, and pandemics. Most populations in these settings lack access to mental health services.

- In prior studies, people who are not mental health specialists have been trained to effectively deliver psychological treatments, including Problem Management Plus (PM+), which is a brief 5-session intervention. However, there has only been one prior study of a group-based format of PM+ delivered by nonspecialists.

- As the use of nonspecialists increases, there are new questions about how these psychological interventions work when delivered by someone who is not a mental health professional. Studying potential mechanisms of action (i.e., how the intervention works) could be instrumental to increasing effectiveness when scaling up.

### What did the researchers do and find?

- The researchers evaluated the effectiveness of Group PM+ in a setting prone to humanitarian emergencies with delivery of the intervention by briefly trained nonspecialists, and the researchers evaluated a potential mechanism of action involved in reducing psychological distress.

- Adults who received Group PM+ delivered by nonspecialists in their communities showed greater reduction in psychological distress at 3 months after the intervention compared to adults who had been offered referral options for services. At 3 months after the intervention, approximately 1 out of 3 adults in Group PM+ had a 50% reduction in depression symptoms compared to 1 out of 6 who were in the control arm receiving referral options for services.

- Regarding how the intervention produced changes, a third of the differences between study arms was explained by participants' use of psychosocial skills taught in Group PM+, such as breathing exercises, problem-solving techniques, and seeking social support.

### What do these findings mean?

- In humanitarian emergencies with a lack of mental health specialists, a 5-session group psychological treatment delivered by nonspecialists can be used to modestly reduce psychological distress and depression symptoms.

- The benefits of Group PM+ are partly explained by the degree of psychosocial skill use being promoted through the intervention. Therefore, future use of the intervention should explore how to enhance practice of these skills in daily life.

- Further research is needed to evaluate how the impact of Group PM+ differs by gender because the study points toward potentially less benefit among men compared to women.

## Introduction

Globally, 235 million people are impacted by humanitarian emergencies [1]. With the Coronavirus Disease 2019 (COVID-19) pandemic's impacts on healthcare, livelihoods, education, and

security, more populations will experience humanitarian emergencies and associated mental health problems [2]. For most populations in humanitarian emergencies, the burden of mental health problems outweighs the availability of mental health services, and the number of mental health specialists is not sufficient to care for all persons in need. Increasingly, there is evidence that persons without a professional mental health education can effectively deliver psychological interventions [3].

Problem Management Plus (PM+) is a 5-session transdiagnostic intervention that incorporates multiple therapeutic techniques and is designed for delivery by nonspecialists in humanitarian settings [4]. PM+, delivered in an individual format, has shown benefit in Pakistan [5] and Kenya [6], and a group version (Group PM+), also consisting of 5 sessions, has shown benefit among women in Pakistan [7]. As PM+ is increasingly used globally, including in the United States in response to the COVID-19 pandemic [8], one of the key questions is determining the mechanisms of action by which benefits are achieved so these mechanisms can be emphasized when adapting and implementing in new settings [9]. Therefore, in addition to this being the second trial of Group PM+, it is the first trial of PM+ evaluating mechanisms of action. This is also the first Group PM+ trial including both women's and men's groups.

We conducted a 2-arm, single-blind cluster randomized controlled trial (cRCT) that compared Group PM+ and enhanced usual care (EUC) among participants with psychological distress and functional impairment in Nepal, a country prone to humanitarian emergencies and their negative mental health sequelae [10–12]. Outcomes were independently assessed at baseline, midline (7 weeks post-baseline, which was approximately 1 week after treatment in the experimental arm), and endline (20 weeks post-baseline, which was approximately 3 months posttreatment). We hypothesized that at 3-month follow-up, Group PM+ would result in lower psychological distress scores (primary hypothesis), as well as fewer depression symptoms, posttraumatic stress disorder (PTSD) symptoms, and somatic complaints (secondary hypotheses) relative to EUC, at the individual participant level. We hypothesized that higher levels of psychosocial skill use (the proposed mechanisms of action) will mediate treatment outcomes. The trial protocol contains full design details [13].

## Methods

### Setting

In rural settings in Nepal, specialized mental healthcare is largely absent [14]. The study was conducted in Morang district, in eastern Nepal. Morang's population is mixed by caste and ethnicity, with Nepali language being spoken by the majority. Annual floods affect significant parts of the district [15]. The region was impacted by a civil war from 1996 to 2006.

### Participants

Participants were at least 18 years of age and could understand and speak Nepali. Eligibility criteria were current psychological distress and impaired functioning. Current psychological distress was assessed with categorical endorsement (yes/no) of a local idiom of distress ("heart–mind problems," Nepali: *manko samasya*) [16], which has 94% sensitivity for structured clinical depression diagnoses in Nepal [17]. Functional impairment was determined with the World Health Organization Disability Assessment Schedule II (WHODAS-II) [18], for scores >16. Exclusion criteria were presence of a severe mental disorder (e.g., psychosis) assessed through a comparison with a vignette [19], cognitive impairment assessed with a disabilities questionnaire [20], or harmful alcohol use (determined by a score >16 on the Alcohol

Use Disorders Identification Test, AUDIT) [21]. Participants were categorized by self-identified gender rather than sex at birth. This was because treatment groups were formed separately based on gender (i.e., women's or men's groups), and it was considered inappropriate to expect participants to join groups that were based on sex at birth when that was discordant with their social and community gender identity. No changes were made to eligibility criteria after trial commencement.

## Intervention

Group PM+ was developed by the World Health Organization (WHO), and the manual is publicly available [4,22]. Group PM+ is delivered in 5 weekly sessions lasting approximately 2.5 hours each. Group PM+ comprises the following evidence-based techniques: (a) problem-solving; (b) stress management through deep breathing; (c) behavioral activation; and (d) promoting social support. The Group PM+ training manual and implementation materials were adapted for Nepal using the mental health Cultural Adaptation and Contextualization for Implementation (mhCACI) procedure [23]. The facilitators of Group PM+ were not mental health specialists. These nonspecialists were recruited following a set of predefined criteria, including living in the communities where the project took place, not having received prior mental health training, having completed higher secondary school (equivalent to high school graduation), as well as based on interviews demonstrating adequate communication skills and motivation to serve members of their communities. The facilitators were compensated as full-time employees of the implementing nongovernmental organization, Transcultural Psychosocial Organization (TPO) Nepal, during the training and Group PM+ delivery period. Facilitators first received a 10-day training on foundational helping skills [24], followed by 10 days of Group PM+ facilitator training with subsequent supervised practice sessions. Face-to-face group supervision was provided weekly. The 2 supervisors were staff members of TPO Nepal. Both were trained psychologists and experienced counselors. In this trial, Group PM+ was delivered at the cluster level (1 group was formed per ward). Group PM+ sessions were held in easily accessible community locations (e.g., primary healthcare centers, community centers, and civil society organization offices).

The EUC control arm and Group PM+ arm participants received a time-restricted (approximately 30 minutes) family psychoeducation meeting conducted by briefly trained local community members, consisting of (a) basic information on adversity and mental health; and (b) information about referral options to primary care providers trained in WHO mental health Gap Action Programme-Intervention Guide (mhGAP-IG) [25]. The family psychoeducation meeting and referral information were the only additional services provided to EUC participants outside of what was normally available to the general population.

## Randomization and masking

The cluster unit of randomization was the ward, the smallest administrative unit in Nepal. Of eligible wards, 20% were allocated for men's groups and 80% for women's groups. This gender ratio was based on service use in a prior district-wide mental health program [26]. We followed a restricted randomization procedure. We first stratified by ward gender and then implemented covariate constrained randomization to account for 3 binary cluster-level covariates: (a) access to existing mental health services ("close" <1 hour to reach services); (b) disaster risk ("high frequency" of landslides or flooding, ≥ once in the past 3 years); and (c) rural/urban status. Wards with mainly non-Nepali speaking inhabitants were excluded.

The research team (research assistants administering all interviews, research supervisors, and study statisticians) were masked to allocation. We limited risk of unmasking by utilizing a

separation between assessors and Group PM+ facilitators (e.g., using 2 separate offices) and by prompting research participants not to share information with the assessors on the type of intervention that they received. To assess attainment of adequate masking, research assistants were asked to guess the allocation status of study participants after each interview. Unblinding was defined as a research assistant correctly identifying the participant's study arm after midline and before endline (the primary endpoint).

## Procedures

Recruitment of participants happened after randomization of clusters. We recruited and trained 1 or 2 community members per ward, who recruited people perceived to have "heart–mind problems," using the Community Informant Detection Tool (CIDT) [27]. The CIDT is a vignette-based tool for proactive case detection, developed and evaluated in Nepal, with good positive predictive value for depression when compared to structured clinical interviews [19]. In the potential participant's home or another location of the participant's choosing (e.g., local health facility), research assistants obtained individual consent for screening, then screened participants for eligibility. All eligible participants then received family meetings in the participant's home. Subsequently, research assistants conducted baseline interviews in the participant's home or other preferred location. Prior to conducting this cRCT, we completed a Group PM+ pilot study to test trial procedures, in which all predefined feasibility and acceptability criteria were met (e.g., recruitment and retention milestones, treatment fidelity, and few adverse events) [28]. The trial was ended as planned after completion of follow-up.

## Instruments

The primary outcome is psychological distress at the individual participant level, measured using the General Health Questionnaire (GHQ-12) [29], which has been validated in Nepal [30]. Secondary outcomes include depression symptoms measured using the Patient Health Questionnaire (PHQ-9) [31], also validated in Nepal [17]; "heart–mind problems" [17]; general functioning measured with WHODAS-II [26]; PTSD symptoms using an adapted 8-item Nepali version of the PTSD CheckList (PCL) [32,33] based on longer versions previously used in Nepal [10,12,34,35]; perceived social support using the Multi-dimensional Scale of Perceived Social Support (MSPSS) [36,37]; and the Somatic Symptom Scale 8 (SSS-8) [38]. No changes were made to trial outcomes after the trial commenced.

Demographic characteristics of participants, traumatic events [35,39], and exposure to natural disasters were recorded at baseline. We developed a 10-item Reducing Tension Checklist (RTC) as a measure of the intervention mechanism of action (see RTC in **S1 Text**). RTC measures the use of behavioral and psychosocial coping skills related to Group PM+ content, aiming to evaluate skill acquisition relevant for Group PM+. Each of the 4 active ingredients of Group PM+ (stress management through deep breathing, problem-solving, behavioral activation, and seeking social support) have multiple items in the RTC, with a total score indicating the level of combined skill acquisition (score range = 0 to 40). As the RTC was used among both Group PM+ and EUC participants, it is worded so that it is relevant for both groups.

Psychological treatment competency of the facilitators was evaluated during training with the ENhancing Assessment of Common Therapeutic factors (ENACT) rating of standardized role plays, which was developed in Nepal and has been used in a range of humanitarian settings [40–42]. During implementation, fidelity to Group PM+ was assessed with a tool adapted from components in the PM+ manual. Fidelity and competency were assessed by supervisors observing 2 sessions per treatment group using standardized checklists.

## Analysis

All analyses reflect the clustered longitudinal nature of the outcome data. Analyses are described in the published protocol [13] and the statistical analysis plan, which was signed before unmasking the study. Primary analyses used the "intention-to-treat" population. Sub-group analyses excluded intervention arm participants who attended fewer than 4 Group PM + sessions ("non-completers") while using data from all control arm participants. No interim analyses were planned. Results are reported in accordance with the cluster RCT Consolidated Standards of Reporting Trials (CONSORT) extension [43] (see **S2 Text** with CONSORT checklist, cluster trials extension version).

This cRCT was designed to have at least 90% power to detect a moderate effect size of 0.46 for the primary outcome (GHQ-12) at the primary time point (endline). Assumptions were the following: intracluster correlation coefficient (ICC) due to clustering by ward of 0.2 within each arm (based on population level data from a community-based sample in Nepal [26]), 2-tailed 5% significance level, 72 clusters (36 per arm) with 8 participants per ward, and drop-out of up to 2 per ward.

The midline and endline measures of each outcome were jointly modeled using a linear mixed effects model with ward-level predictors of arm, time, arm by time, ward gender, access to mental health services, disaster risk and rural/urban status, as well as the participant-level baseline measure of the outcome. Random intercepts were included for participant and ward, with different ward-level ICC for each treatment arm. Sensitivity analyses additionally pre-specified adjustment for predictors of missing outcomes (where we note that, in practice, none were identified). Secondary binary outcomes were analyzed using the same predictors within the modified Poisson framework to obtain both risk ratios and risk differences [44]. Exploratory subgroup analyses assessed if there were different treatment effects according to gender and baseline depressive symptoms by separately including each of the 2 variables in the analysis model together with interactions between that variable and intervention arm and time point. Using the group–mean centering approach of Zhang and colleagues [45], mediation of the intervention effect on the primary outcome at endline was evaluated using a difference-in-coefficients mediation framework for cRCTs to estimate both between- and within-ward effects for the hypothesized participant-level mediator of RTC skill use scores (i.e., number of Group PM+ strategies used) at midline. Specifically, the group–mean centering approach requires that 2 models are fitted. First, endline GHQ-12 was regressed on the following covariates: arm, ward gender (to account for the stratified design), the 3 covariates used in the constrained randomization procedure (access to mental health services, disaster risk, and rural/urban status), and each participant's baseline measures of both RTC and of GHQ. A random intercept for ward was included with different ICCs for each arm. A second model was fitted with 2 additional variables: the ward-level mean RTC and each participant's RTC deviation from the ward mean. From these 2 models, the mediated effect was estimated as per Zhang and colleagues [45].

## Ethics

The trial is registered with ClinicalTrials.gov (NCT03747055). The study has been approved by the Nepal Health Research Council, Kathmandu, Nepal (Ref 481, September 2018) and WHO Ethical Review Committee (version 3, ID 2817, October 2018). Because of low literacy levels, all participants were read the consent form by the research assistant and given an opportunity to discuss questions about the study with the research assistant. No changes were made to methods or procedures after trial commencement.

## Results

### Participant flow and recruitment

Out of 100 wards assessed for eligibility, 72 were eligible wards. A total of 58 wards were selected for female participants and 14 wards for male participants (see **Fig 1**). The wards were randomized to the EUC or Group PM+ arms. Participant recruitment occurred from November 25, 2018 to May 28, 2019 (final follow-up of all participants was completed September 30, 2019). In the control wards, 1,169 adults were screened, and 324 met eligibility criteria. In the Group PM+ wards, out of 885 persons screened, 319 met eligibility criteria (see **Tables A and B in S3 Text**).

See **Table 1** for a description of the sample. Additional baseline demographic variables including exposure to disasters and traumatic events and demographics by gender events are provided in **Table C in S3 Text**. Reasonable balance between arms was observed for most

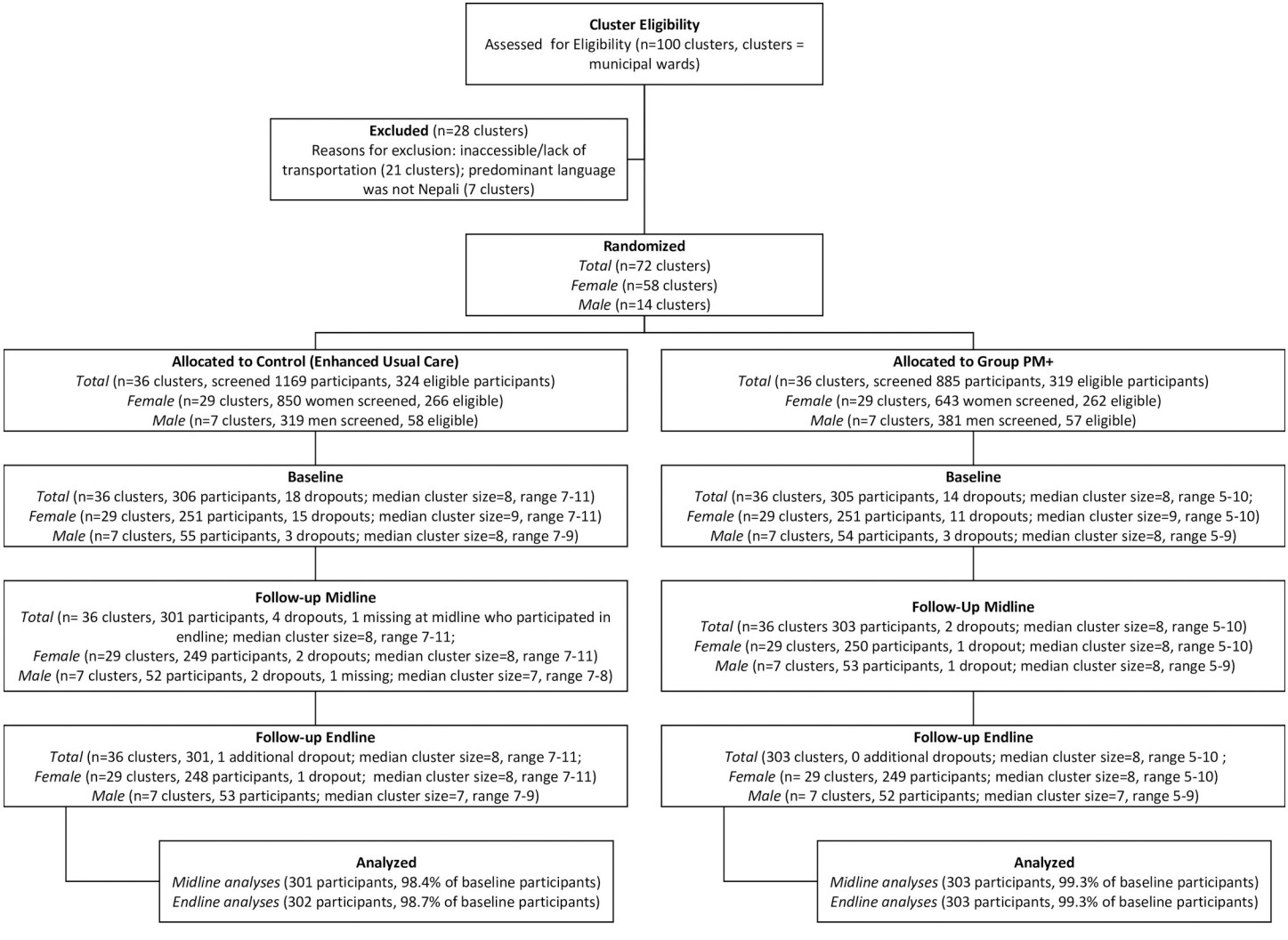

**Fig 1. CONSORT flowchart for Group PM+ cRCT in community settings Morang, Nepal, conducted November 25, 2018 through September 30, 2019.** Midline is 7 weeks post-baseline (after completion of the intervention in the Group PM+ arm). Endline is 20 weeks post-baseline (approximately 3 months after completion of the intervention in the Group PM+ arm). Group PM+ consists of 5 weekly group therapy sessions. EUC is a brief (30 minutes) family psychoeducation session and passive referrals to primary care–based mental health services. CONSORT, Consolidated Standards of Reporting Trials; cRCT, cluster randomized controlled trial; EUC, enhanced usual care; PM+, Problem Management Plus.

**Table 1. Baseline characteristics by arm.**

| Baseline characteristics | Control | Group PM+ | Total |
|---|---|---|---|
| | (*N* = 306) | (*N* = 305) | (*N* = 611) |
| **Age (years)** | | | |
| Mean (SD) | 44.1 (14.0) | 45.5 (14.8) | 44.8 (14.4) |
| Median (Q1, Q3) | 45.0 (33.0, 54.0) | 44.0 (35.0, 55.0) | 45.0 (34.0, 55.0) |
| Min, max | 18.0, 83.0 | 18.0, 91.0 | 18.0, 91.0 |
| **Gender** | | | |
| Female | 251 (82.0%) | 251 (82.3%) | 502 (82.2%) |
| **Education** | | | |
| Cannot read or write | 88 (28.8%) | 88 (28.9%) | 176 (28.8%) |
| Literate or informal education | 84 (27.5%) | 80 (26.2%) | 164 (26.8%) |
| Primary level | 72 (23.5%) | 77 (25.2%) | 149 (24.4%) |
| Secondary | 47 (15.4%) | 45 (14.8%) | 92 (15.1%) |
| Higher secondary | 13 (4.2%) | 13 (4.3%) | 26 (4.3%) |
| University | 2 (0.7%) | 2 (0.7%) | 4 (0.7%) |
| **Occupation** | | | |
| Farmer | 104 (34.0%) | 93 (30.5%) | 197 (32.2%) |
| Business or job | 33 (10.7%) | 34 (11.1%) | 67 (10.9%) |
| Daily wage laborer | 35 (11.4%) | 33 (10.8%) | 68 (11.1%) |
| Unemployed | 9 (2.9%) | 14 (4.6%) | 23 (3.8%) |
| Student | 8 (2.6%) | 4 (1.3%) | 12 (2.0%) |
| Housewife | 113 (36.9%) | 120 (39.3%) | 233 (38.1%) |
| Other | 4 (1.3%) | 7 (2.3%) | 11 (1.8%) |
| **Caste categories** | | | |
| Upper caste (Brahman, Chhetri) | 110 (36.0%) | 110 (36.1%) | 220 (36.0%) |
| Janajati | 78 (25.5%) | 73 (23.9%) | 151 (24.7%) |
| Madhesi and Local Indigenous | 48 (15.6%) | 57 (18.6%) | 105 (17.1%) |
| Other | 70 (22.8%) | 65 (21.3%) | 135 (22.0%) |
| **Religion** | | | |
| Hindu | 257 (84.0%) | 267 (87.5%) | 524 (85.8%) |
| Buddhist | 10 (3.3%) | 8 (2.6%) | 18 (2.9%) |
| Muslim | 1 (0.3%) | 0 (0.0%) | 1 (0.2%) |
| Christian | 19 (6.2%) | 17 (5.6%) | 36 (5.9%) |
| No religion | 1 (0.3%) | 1 (0.3%) | 2 (0.3%) |
| Other | 18 (5.9%) | 12 (3.9%) | 30 (4.9%) |
| **Marital status** | | | |
| Unmarried | 16 (5.2%) | 17 (5.6%) | 33 (5.4%) |
| Married | 249 (81.4%) | 242 (79.3%) | 491 (80.4%) |
| Widowed | 27 (8.8%) | 39 (12.8%) | 66 (10.8%) |
| Divorced | 3 (1.0%) | 3 (1.0%) | 6 (1.0%) |
| Separated | 11 (3.6%) | 4 (1.3%) | 15 (2.5%) |
| **Primary language** | | | |
| Nepali | 252 (82.4%) | 249 (81.6%) | 501 (82.0%) |
| Other | 54 (17.6%) | 56 (18.4%) | 110 (18%) |
| **Household size** | | | |
| Living alone | 11 (3.6%) | 9 (3.0%) | 20 (3.3%) |
| With 1 other person | 29 (9.5%) | 29 (9.5%) | 58 (9.5%) |
| With 2 to 3 other people | 125 (40.8%) | 122 (40.0%) | 247 (40.4%) |

(*Continued*)

**Table 1.** (Continued)

| Baseline characteristics | Control | Group PM+ | Total |
|---|---|---|---|
| | (N = 306) | (N = 305) | (N = 611) |
| With 4 or more other people | 141 (46.1%) | 145 (47.5%) | 286 (46.8%) |
| **Chronic diseases** | | | |
| Yes | 94 (30.7%) | 96 (31.5%) | 190 (31.1%) |
| **If yes to chronic disease** | | | |
| Cancer | 2 (2.1%) | 3 (3.1%) | 5 (2.6%) |
| Diabetes | 17 (18.1%) | 18 (18.8%) | 35 (18.4%) |
| Hypertension | 36 (38.3%) | 36 (37.5%) | 72 (37.9%) |
| Asthma | 14 (14.9%) | 21 (21.9%) | 35 (18.4%) |
| Other | 25 (26.6%) | 18 (18.8%) | 43 (22.6%) |
| **Who do you live with?** | | | |
| Extended family with spouse | 101 (33.0%) | 81 (26.6%) | 182 (29.8%) |
| Extended family without spouse | 15 (4.9%) | 23 (7.5%) | 38 (6.2%) |
| With parents | 10 (3.3%) | 9 (3.0%) | 19 (3.1%) |
| Maternal home (Nepali: Maiti) | 6 (2.0%) | 5 (1.6%) | 11 (1.8%) |
| Spouse only | 20 (6.5%) | 17 (5.6%) | 37 (6.1%) |
| Spouse and children only | 107 (35.0%) | 116 (38.0%) | 223 (36.5%) |
| Other | 47 (15.4%) | 54 (17.7%) | 101 (16.5%) |
| **Indicators of economic status (yes)** | | | |
| Concrete building | 39 (12.7%) | 48 (15.7%) | 87 (14.2%) |
| Electricity | 271 (88.6%) | 277 (90.8%) | 548 (89.7%) |
| Drinking water | 276 (90.2%) | 270 (88.5%) | 546 (89.4%) |
| Radio | 85 (27.8%) | 78 (25.6%) | 163 (26.7%) |
| Television | 186 (60.8%) | 190 (62.3%) | 376 (61.5%) |
| Simple mobile phone | 252 (82.4%) | 242 (79.3%) | 494 (80.9%) |
| Smart mobile phone | 151 (49.3%) | 163 (54.5%) | 314 (51.9%) |
| Bicycle | 202 (66.0%) | 226 (74.1%) | 428 (70.0%) |
| LP gas | 224 (73.2%) | 231 (75.7%) | 455 (74.5%) |
| **Ever taken medication for mental health problems** | | | |
| No | 263 (85.9%) | 287 (94.1%) | 550 (90.0%) |
| Yes | 31 (10.1%) | 10 (3.3%) | 41 (6.7%) |
| Do not know | 12 (3.9%) | 8 (2.6%) | 20 (3.8%) |
| **Ever received counseling services (e.g., counselor, doctor, or religious advisor)** | | | |
| 0 time | 291 (95.1%) | 297 (97.4%) | 588 (96.2%) |
| 1 to 4 times | 7 (2.3%) | 2 (0.7%) | 9 (1.5%) |
| 5 to 10 times | 6 (1.9%) | 4 (1.3%) | 10 (1.6%) |
| >10 times | 2 (0.7%) | 2 (0.7%) | 4 (0.7%) |
| **Traumatic and natural disaster exposures** | | | |
| Ever experienced a natural disaster (yes) | 163 (53.3%) | 145 (47.5%) | 308 (50.4%) |
| When did the natural disaster occur? | | | |
| 0 to 3 years ago | 22 (13.5%) | 31 (21.4%) | 53 (17.2%) |
| 3 years ago | 141 (86.5%) | 114 (78.6%) | 255 (82.8%) |
| Been in a serious accident | 65 (21.2%) | 51 (16.7%) | 116 (19.0%) |
| Had a serious sickness | 227 (74.2%) | 216 (70.8%) | 443 (72.5%) |
| Been in the military or war zone | 26 (8.5%) | 20 (6.6%) | 46 (7.5%) |
| Seen/had a death/murder of close family or friend | 63 (20.6%) | 64 (21.0%) | 127 (20.8%) |
| Seen/had close friend/family member commit suicide | 114 (37.3%) | 91 (29.8%) | 205 (33.6%) |

*(Continued)*

**Table 1.** (Continued)

| Baseline characteristics | Control | Group PM+ | Total |
|---|---|---|---|
| | (*N* = 306) | (*N* = 305) | (*N* = 611) |
| Been attacked with a gun/knife | 30 (9.8%) | 37 (12.1%) | 67 (11.0%) |
| Been attacked without weapon | 40 (13.1%) | 48 (15.7%) | 88 (14.4%) |
| Beaten as a child | 77 (25.2%) | 73 (23.9%) | 150 (24.5%) |
| Had adult sexual contact before age 13 | 7 (2.3%) | 8 (2.6%) | 15 (2.5%) |
| Had unwanted sexual contact after age 13 | 22 (7.2%) | 25 (8.2%) | 47 (7.7%) |

PM+, Problem Management Plus.

baseline demographic variables. As per CONSORT recommendations, we did not obtain *p*-values for these comparisons. Likewise, reasonable balance was also observed for the population of completers (see **Table D in S3 Text**).

## Treatment exposure and enhanced usual care

Median competency of the 12 facilitators in common factors as measured with ENACT after training was 81% (range 61% to 100%; see **Table 2**). Median fidelity during Group PM+ delivery was 2.8 (minimum = 2.5, maximum = 3, on a scale of 0 to 3; see **Table E in S3 Text**). In the Group PM+ arm, 238 (78%) participants completed treatment, defined as attending 4 (*N* = 72) or 5 sessions (*N* = 166) (**Table F in S3 Text**). **Table G in S3 Text** is an overview of other services received by participants during the study period (e.g., traditional healing, medication, and other counseling services).

## Primary outcome

In intention-to-treat analyses, the Group PM+ arm was associated with lower GHQ-12 scores at both midline and endline compared to the control arm (see **Table 3** and **Table H in S3**

**Table 2. Competency and fidelity of Group PM+ facilitators.**

| Facilitator | Pretraining competency in common factors[1] | Post-training competency in common factors[1] | Fidelity per session to Group PM+[2], median (range: min, max) |
|---|---|---|---|
| 1 | 72% | 100% | 2.9 (2.6, 3.0) |
| 2 | 22% | 100% | 2.9 (2.8, 3.0) |
| 3 | 0% | 100% | 3.0 (2.7, 3.0) |
| 4 | 44% | 89% | 2.9 (2.9, 3.0) |
| 5 | 44% | 67% | 2.5 (2.5, 2.6) |
| 6 | 44% | 61% | 2.8 (2.7, 2.8) |
| 7 | 39% | 72% | 2.7 (2.6, 2.7) |
| 8 | 83% | 100% | 2.8 (2.8, 2.9) |
| 9 | 56% | 78% | 2.8 (2.6, 2.8) |
| 10 | 44% | 72% | 2.8 (2.7, 2.8) |
| 11 | 33% | 78% | 2.8 (2.8, 2.8) |
| 12 | 50% | 83% | 2.8 (2.5, 3.0) |

[1]Competency evaluated with observed structured role plays with standardized client actors, assessed with the ENACT, range is 0% to 100% with higher scores reflecting competency on more skills, with a total of 18 skills.

[2]Fidelity assessed with session specific Group PM+ checklist completed by supervisors observing actual sessions with participants. All facilitators were observed for at least 2 group sessions, range is 0 to 3, with higher scores reflecting greater fidelity.

ENACT, ENhancing Assessment of Common Therapeutic factors; PM+, Problem Management Plus.

**Table 3. Intervention effects (mean differences and 95% confidence intervals) on primary outcome: psychological distress measured with the GHQ-12.**

| Primary outcome (GHQ-12) | Mean (SD), n | | | Estimated treatment effect | | | Variance components[e] | | | |
| --- | --- | --- | --- | --- | --- | --- | --- | --- | --- | --- |
| | Control (N = 306) | PM+ (N = 305) | Total (N = 611) | Mean difference (95% CI) | p-Value | SMD (95% CI)[f] | Cluster (residential ward) | | Person | Residual |
| | | | | | | | Control | PM+ | | |
| **ITT[a]** | | | | | | | | | | |
| Baseline | 20.9 (6.0), n = 306 | 21.2 (6.3), n = 305 | 21.0 (6.1), n = 611 | - | - | - | - | - | - | - |
| Midline | 20.3 (6.6), n = 301 | 17.7 (6.9), n = 303 | 19.0 (6.9), n = 604 | −2.7 (−3.7, −1.7) | <0.001 | −0.4 (−0.5, −0.2) | <0.001 | 2.0 | 13.9 | 17.8 |
| Endline | 19.3 (6.5), n = 301 | 18.1 (7.0), n = 301 | 18.7 (6.7), n = 602 | −1.4 (−2.5, −0.3) | 0.014 | −0.2 (−0.4, −0.0) | 1.6 | 2.4 | | |
| **ITT sensitivity[b]** | | | | | | | | | | |
| Baseline | 20.9 (6.0), n = 306 | 21.2 (6.3), n = 305 | 21.0 (6.1), n = 611 | - | - | - | - | - | - | - |
| Midline | 20.3 (6.6), n = 301 | 17.7 (6.9), n = 303 | 19.0 (6.9), n = 604 | −2.9 (−4.0, −1.9) | <0.001 | −0.4 (−0.6, −0.3) | <0.001 | 2.4 | 12.2 | 17.7 |
| Endline | 19.3 (6.5), n = 301 | 18.1 (7.0), n = 301 | 18.7 (6.7), n = 602 | −1.6 (−2.8, −0.5) | 0.005 | −0.2 (−0.4, −0.1) | 2.5 | 2.3 | | |
| **ITT female[c]** | | | | | | | | | | |
| Baseline | 20.9 (5.9), n = 251 | 20.8 (6.0), n = 251 | 20.8 (5.9), n = 502 | - | - | - | - | - | - | - |
| Midline | 20.6 (6.5), n = 249 | 17.3 (6.8), n = 250 | 18.9 (6.8), n = 499 | −3.0 (−4.1, −1.9) | <0.001 | −0.4 (−0.6, −0.3) | <0.001 | 1.9 | 13.9 | 17.8 |
| Endline | 19.6 (6.5), n = 248 | 18.0 (6.5), n = 249 | 18.8 (6.6), n = 497 | −1.5 (−2.8, −0.3) | 0.017 | −0.2 (−0.4, −0.0) | 1.7 | 2.6 | | |
| **ITT male[c]** | | | | | | | | | | |
| Baseline | 20.9 (6.4), n = 55 | 23.3 (7.2), n = 54 | 22.1 (6.9), n = 109 | - | - | - | - | - | - | - |
| Midline | 19.1 (7.1), n = 52 | 19.5 (7.3), n = 53 | 19.3 (7.2), n = 105 | −1.1 (−3.5, 1.3) | 0.359 | −0.2 (−0.5, 0.2) | <0.001 | 1.9 | 13.9 | 17.8 |
| Endline | 18.1 (6.3), n = 53 | 18.7 (8.7), n = 52 | 18.4, (7.6), n = 105 | −0.9 (−3.5, 1.8) | 0.529 | −0.1 (−0.5, 0.2) | 1.7 | 2.6 | | |
| **Completers[d]** | | | | | | | | | | |
| Baseline | 20.9 (6.0), n = 306 | 21.1 (6.4), n = 238 | 21.0 (6.2), n = 544 | - | - | - | - | - | - | - |
| Midline | 20.3 (6.6), n = 301 | 17.3 (6.7), n = 238 | 19.0 (6.8), n = 539 | −3.0 (−4.1, −1.9) | <0.001 | −0.4 (−0.6, −0.3) | <0.001 | 2.7 | 12.7 | 17.9 |
| Endline | 19.3 (6.5), n = 301 | 18.1 (6.8), n = 237 | 18.8 (6.7), n = 538 | −1.4 (−2.6, −0.3) | 0.015 | −0.2 (−0.4, −0.0) | 1.7 | 2.6 | | |

[a]Primary analytic model (see Methods).

[b]Primary analytic model with 5 additional baseline covariates: age categories, caste categories, exposure to natural disasters, baseline WHODAS, and baseline PHQ-9.

[c]Gender effects obtained by adding 3 interactions to primary analytic model: arm by gender, time by gender, and arm by time by gender.

[d]Primary analytic model applied to the completer population.

[e]In units$^2$. Can be used to obtain ICC for each treatment group at each time point using arm time-specific variance/residual variance. ICC estimates for ITT the analysis are <0.001 and 0.082 for control at midline and endline, respectively, and 0.101 and 0.119 for PM+ at midline and endline, respectively.

[f]SMDs calculated by dividing the estimated mean difference by the overall standard deviation across the 2 arms combined.

CI, confidence interval; GHQ-12, General Health Questionnaire; ICC, intraclass correlation coefficient; ITT, intention to treat; PHQ-9, Patient Health Questionnaire; PM+, Problem Management Plus; SMD, standardized mean difference; WHODAS, World Health Organization Disability Assessment Schedule.

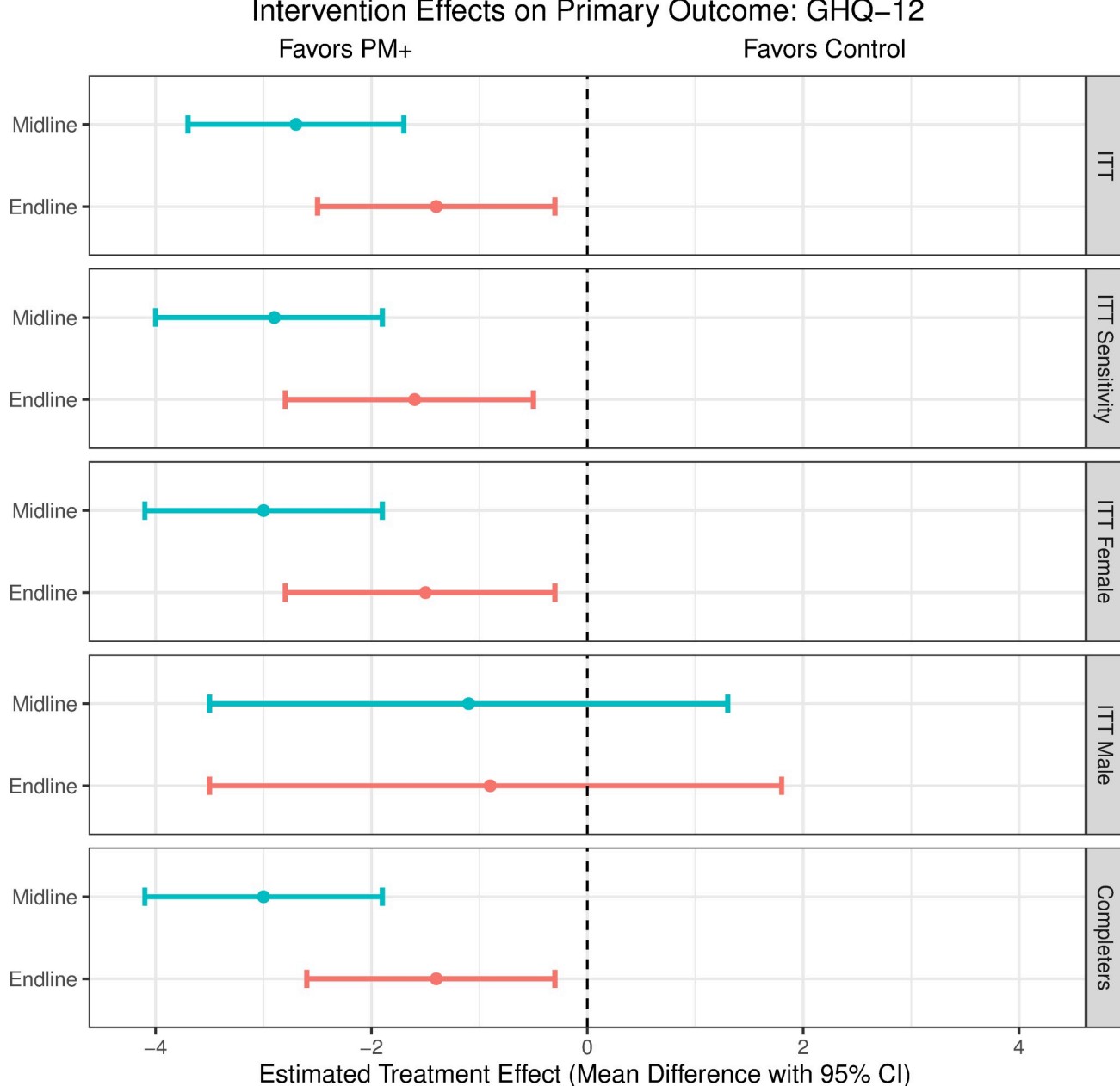

**Fig 2. Intervention effects on primary outcome (GHQ-12) at midline (approximately 1 week after treatment) and endline (approximately 3 months after treatment).** GHQ-12, General Health Questionnaire; ITT, intention to treat; PM+, Problem Management Plus.

**Text**). At 3 months, this effect is significant at the 5% level with an estimated GHQ-12 score of 1.4 units (95% CI: 0.3, 2.5; $p = 0.014$) lower in Group PM+ compared to control when adjusting for the baseline GHQ-12, standardized mean difference (SMD) = −0.2 (95% CI: −0.4, −0.0) (see **Fig 2**). Using the primary analytic model, the ICC for the control arm was 0.00 at midline and 0.08 at endline, and, for the Group PM+ arm, 0.10 and 0.12. When additional prespecified covariates were adjusted for (e.g., age, caste, exposure to disasters, baseline WHODAS, and baseline PHQ-9), similar results were obtained. Likewise, similar results were obtained for the completers population (**Table 3** and **Table I in S3 Text**). There were minimal missing

outcome data (98.8% completed endline). There was no indication of baseline covariates differing by study arm and midline or endline data availability so we did not perform other sensitivity analyses to account for missing outcome data (**Tables J and K in** S3 Text). There was no evidence of an interaction between gender and the treatment effect. In gender-specific subanalyses, there was a smaller estimated benefit of Group PM+ for men and 95% confidence intervals spanned a between-arm difference of 0; this is partly explained due to only 20% of the sample being male (**Table 3** and **Table L in** S3 Text).

## Secondary outcomes

The Group PM+ arm was associated with lower depression symptoms at 3 months posttreatment (PHQ-9 mean difference = −1.0, 95% CI: −1.8, −0.1, $p = 0.028$). Group PM+ was not associated with lower functional impairment (WHODAS mean difference = 1.5, 95% CI: −3.4, 0.4, $p = 0.118$), PTSD symptoms (PTSD CheckList [PCL] mean difference = −1.0, 95% CI: −2.2, 0.1, $p = 0.084$), perceived social support (MSPSS mean difference = 1.0, 95% CI: −0.3, 2.3, $p = 0.138$), nor somatic symptoms (SSS mean difference = −1.0, 95% CI: −2.2, 0.2, $p = 0.105$); see **Table 4**. For secondary binary outcomes (**Table 5**), at endline, 29.9% of the Group PM + arm participants showed a 50% reduction in PHQ-9 from baseline compared to 17.3% in the control arm, risk ratio = 1.7 (95% CI: 1.2, 2.4, $p = 0.002$). Similarly, 58.8% of participants in the Group PM+ arm had "heart–mind problems" at endline compared to 69.4% of participants in the control arm, risk ratio = 0.8 (95% CI: 0.7, 1.0, $p = 0.042$).

## Mediation analyses

After adjustment for chance imbalance in baseline levels of the mediator (skill use measured with the RTC), the hypothesized mediator was greater in the Group PM+ arm at midline (RTC mean difference = 2.1, 95% CI: 1.0, 3.2, $p < 0.001$) and endline (RTC mean difference = 1.5, 95% CI: 0.3, 2.7, $p = 0.016$); see **Table 4**. The corresponding estimated standardized effects were slightly smaller than for the primary outcome (**Tables 3 and 4**). The group–mean centering mediation analysis of RTC at midline as a mediator of GHQ-12 at endline shows an estimated mediation effect of −0.4 units relative to the estimated intervention effect of −1.3 units (see **Fig 3**) so that the estimated relative portion of the Group PM+ effect on endline GHQ-12 that is mediated by midline RTC is 31% (**Table M in** S3 Text).

## Harms

There were 3 serious adverse events: 1 referral for domestic violence, 1 suicide death in the control arm, and 1 death due to a physical illness in the Group PM+ arm. No harms were attributable to participation in Group PM+ or study trial procedures based on reviews by the Data and Safety Monitoring Board. In all of 3 cases, a trained psychosocial counselor visited the participant or family multiple times, providing grief counseling or other relevant psychosocial support and case management services.

## Unblinding

In order to assess the degree to which research assistants remained blind to the group allocation of the research participants, we asked them to report what type of service they thought each participant they interviewed received. After midline, in the PM+ group, research assistant guessed the wrong type of treatment or reported not to know in 97.4% of the interviews (5.6% were incorrect guesses of individual counseling, 0.7% medication, and 91.1% were "I do not

**Table 4. Intervention effects[a] on secondary and mediator score outcomes (mean differences and 95% confidence intervals for the ITT population).**

| Outcomes | Mean (SD), n | | | Estimated treatment effect | | | Variance components | | | |
|---|---|---|---|---|---|---|---|---|---|---|
| | Control (N = 306) | PM+ (N = 305) | Total (N = 611) | Mean difference (95% CI) | p-Value | SMD (95% CI) | Cluster (residential ward) | | Person | Residual |
| | | | | | | | Control | PM+ | | |
| **Secondary score outcomes** | | | | | | | | | | |
| **WHODAS** | | | | | | | | | | |
| Baseline | 24.6 (6.5), n = 306 | 26.0 (6.8), n = 305 | 25.3 (6.7), n = 611 | - | - | - | - | - | - | - |
| Midline | 19.2 (9.6), n = 301 | 17.1 (8.8), n = 303 | 18.2 (9.3), n = 604 | −3.0 (−4.6, −1.4) | <0.001 | −0.3 (−0.5, −0.2) | 5.7 | 3.5 | 32.6 | 28.2 |
| Endline | 16.5 (8.6), n = 301 | 16.0 (9.6), n = 301 | 16.2 (9.1), n = 602 | −1.5 (−3.4, 0.4) | 0.118 | −0.2 (−0.4, 0.0) | 5.1 | 12.6 | | |
| **PHQ-9** | | | | | | | | | | |
| Baseline | 11.9 (5.0), n = 306 | 12.7 (4.9), n = 305 | 12.3 (5.0), n = 611 | - | - | - | - | - | - | - |
| Midline | 10.9 (5.2), n = 301 | 9.6 (5.0), n = 303 | 10.2 (5.1), n = 604 | −1.7 (−2.5, −0.9) | <0.001 | −0.3 (−0.5, −0.2) | 1.3 | 0.7 | 7.4 | 9.6 |
| Endline | 10.0 (4.4), n = 301 | 9.5 (5.2), n = 301 | 9.8 (4.8), n = 602 | −1.0 (−1.8, −0.1) | 0.028 | −0.2 (−0.4, −0.0) | 0.4 | 2.5 | | |
| **PCL** | | | | | | | | | | |
| Baseline | 21.8 (7.0), n = 306 | 23.0 (6.8), n = 305 | 22.4 (6.9), n = 611 | - | - | - | - | - | - | - |
| Midline | 21.4 (6.9), n = 301 | 20.4 (6.9), n = 303 | 20.9 (6.9), n = 604 | −1.7 (−2.6, −0.8) | <0.001 | −0.2 (−0.4, −0.1) | <0.001 | 0.2 | 13.7 | 15.8 |
| Endline | 20.5 (6.6), n = 301 | 20.2 (7.1), n = 301 | 20.3 (6.9), n = 602 | −1.0 (−2.2, 0.1) | 0.084 | −0.2 (−0.3, 0.0) | 2.3 | 3.6 | | |
| **MSPSS** | | | | | | | | | | |
| Baseline | 30.7 (9.5), n = 306 | 32.5 (9.9), n = 305 | 31.6 (9.7), n = 611 | - | - | - | - | - | - | - |
| Midline | 30.8 (9.5), n = 301 | 32.9 (8.7), n = 303 | 31.9 (9.1), n = 604 | 1.0 (−0.2, 2.1) | 0.097 | 0.1 (−0.0, 0.2) | 2.3 | <0.001 | 17.8 | 25.1 |
| Endline | 31.0 (9.2), n = 301 | 33.2 (9.0), n = 301 | 32.1 (9.2), n = 602 | 1.0 (−0.3, 2.3) | 0.138 | 0.1 (−0.0, 0.2) | 3.1 | 1.9 | | |
| **SSS** | | | | | | | | | | |
| Baseline | 23.0 (6.7), n = 306 | 23.8 (6.9), n = 305 | 23.4 (6.8), n = 611 | - | - | - | - | - | - | - |
| Midline | 23.1 (6.9), n = 301 | 21.5 (6.9), n = 303 | 22.3 (7.0), n = 604 | −2.1 (−3.1, −1.2) | <0.001 | −0.3 (−0.4, −0.2) | 1.1 | 1.4 | 12.7 | 14.5 |
| Endline | 22.1 (7.0), n = 301 | 21.6 (7.5), n = 301 | 21.8 (7.3), n = 602 | −1.0 (−2.2, 0.2) | 0.105 | −0.1 (−0.3, 0.0) | 2.8 | 4.1 | | |
| **Mediator** | | | | | | | | | | |
| **RTC** | | | | | | | | | | |
| Baseline | 26.3 (6.4), n = 306 | 27.5 (6.5), n = 305 | 26.9 (6.5), n = 611 | - | - | - | - | - | - | - |
| Midline | 25.9 (6.2), n = 301 | 28.5 (6.2), n = 303 | 27.2 (6.3), n = 604 | 2.1 (1.0, 3.2) | <0.001 | 0.3 (0.2, 0.5) | 0.3 | 4.8 | 11.3 | 14.8 |
| Endline | 25.5 (5.5), n = 301 | 27.4 (6.3), n = 301 | 26.4 (6.0), n = 602 | 1.5 (0.3, 2.7) | 0.016 | 0.2 (0.0, 0.4) | 1.5 | 5.8 | | |

[a]Based on the primary analytical model.

CI, confidence interval; ITT, intention to treat; MSPSS, Multi-dimensional Scale of Perceived Social Support; PCL, Posttraumatic stress disorder CheckList; PHQ-9, Patient Health Questionnaire; RTC, Reducing Tension Checklist; SMD, standardized mean difference; SSS-8, Somatic Symptom Scale 8; WHODAS, World Health Organization Disability Assessment Schedule.

**Table 5. Intervention effects[a] on secondary binary outcomes (risk ratios and risk differences with 95% confidence intervals for the ITT population).**

| Secondary binary outcomes | n/N (%) | | Estimated treatment effect | | | |
|---|---|---|---|---|---|---|
| | Control | PM+ | Risk ratio (95% CI) | p-Value | Risk difference in percentage points, pp, (95% CI) | p-Value |
| **Heart–mind problems** | | | | | | |
| Baseline | 306/306 (100.0%) | 305/305 (100.0%) | - | - | - | - |
| Midline | 229/301 (76.1%) | 184/303 (60.7%) | 0.8 (0.7, 0.9) | 0.004 | −14.5 pp (−24.4, −4.6) | 0.004 |
| Endline | 209/301 (69.4%) | 177/301 (58.8%) | 0.8 (0.7, 1.0) | 0.042 | −10.2 pp (−20.4, 0.0) | 0.051 |
| **50% reduction in PHQ-9 from baseline** | | | | | | |
| Baseline | - | - | - | - | - | - |
| Midline | 46/301 (15.3%) | 74/303 (24.4%) | 1.6 (1.0, 2.4) | 0.041 | 9.2 pp (0.9, 17.4) | 0.029 |
| Endline | 52/301 (17.3%) | 90/301 (29.9%) | 1.7 (1.2, 2.4) | 0.002 | 12.3 pp (4.1, 20.4) | 0.003 |

[a]Using the modified Poisson approach [44].

CI, confidence interval; ITT, intention to treat; PHQ-9, Patient Health Questionnaire; PM+, Problem Management Plus.

know"); for the EUC arm, the research assistants guessed wrong or reported not to know in 92.0% of the interviews (incorrect guesses included 2.7% individual counseling and 2.3% medication, and 87.0% were "I do not know").

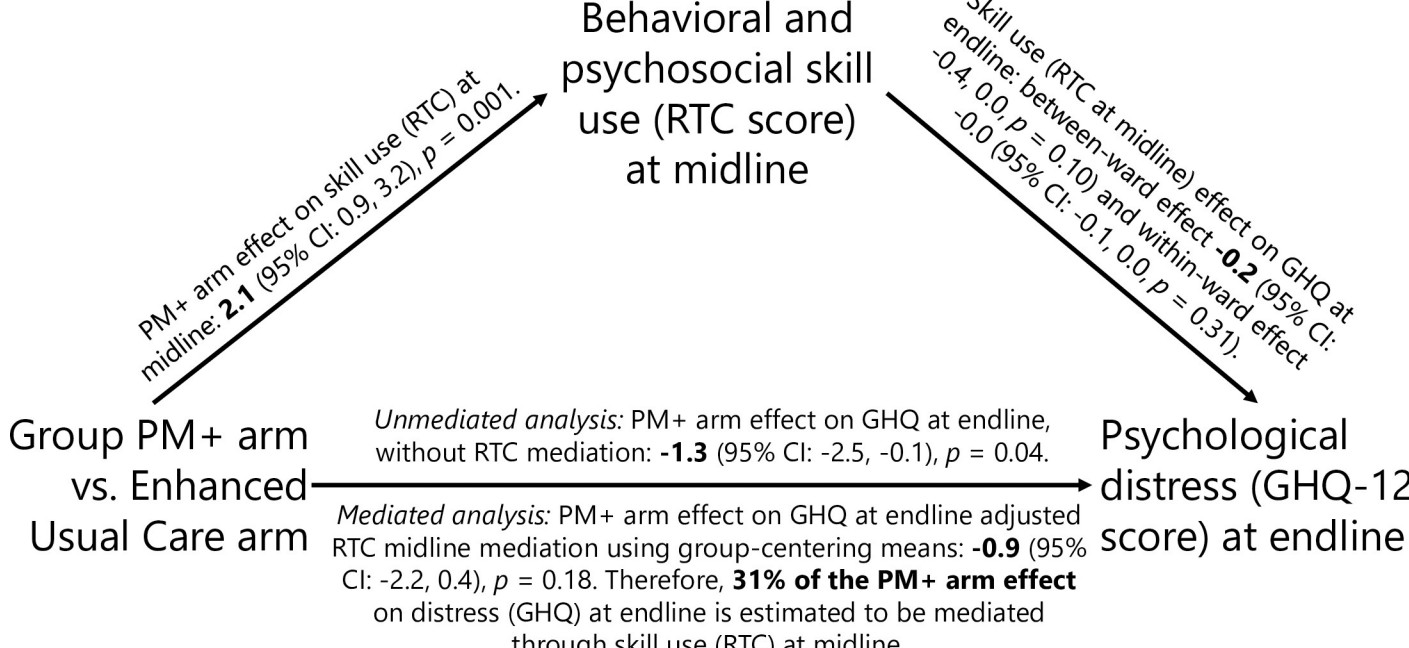

**Fig 3. Hypothesized pathway with model-estimated effects for mediation of behavioral and psychosocial skill use at midline on psychological distress at endline.** Midline is 7 weeks post-baseline (after completion of the intervention in the Group PM+ arm). Endline is 20 weeks post-baseline (approximately 3 months after completion of the intervention in the Group PM+ arm). GHQ-12, General Health Questionnaire; PM+, Problem Management Plus; RTC, Reducing Tension Checklist.

## Discussion

This study evaluated the effectiveness of Group PM+ for people with psychological distress in disaster-prone communities. Results show initially moderate treatment effects (SMD = 0.40 posttreatment) and smaller benefits at 3-month follow-up (SMD = 0.20) in reducing psychological distress. There were benefits in depression symptom reduction at 3-month follow-up (29.9% response rate in Group PM+ compared to 17.3% among controls), which translates to a 70% greater likelihood of reducing symptoms by half due to receiving Group PM+. Similarly, "heart–mind problems" (a local idiom of distress) were present in 58.8% of Group PM+ participants compared to 69.4% of controls at 3-month follow-up. There were no significant between-group differences for other secondary outcomes at endline.

When comparing these results to other psychological treatments studies in low- and middle-income countries, the immediate reduction on psychological distress post-intervention (midline SMD of 0.40) is comparable to the pooled effect for treatments for common mental disorders (SMD = 0.49) [3]. It was lower than the pooled effect for treatments for depression in humanitarian settings (SMD = 0.87) [46]. A cRCT similar to ours evaluating Group PM+ among only women in Pakistan demonstrated larger effect sizes on all outcomes [7].

A unique contribution of this study was evaluating potential mechanism of action: skill use aligned with the treatment's mechanism of action. We found that 31% of the treatment effect in reducing psychological distress at endline is estimated to be mediated by participants' utilization of the therapeutic strategies underlying Group PM+. This suggests that efforts to increase and maintain skill use, such as giving booster sessions, could further enhance the benefit. Studying mechanisms of action may also determine who will most benefit from the intervention. For example, psychosocial and behavioral skills use was already high in this population at baseline (RTC mean of 26.9) compared to the population in the Group PM+ pilot study in a different district in Nepal (RTC baseline of 12.9) [28]. Use of these skills may be greater in settings with higher education levels and greater access to resources. These population characteristics were different between our pilot and full trial site, possibly impacting the magnitude of treatment effect. PM+ may show greater benefits in populations and settings where such skill use is low, which was observed in our pilot site, and benefits may be attenuated in populations with higher preexisting general psychosocial skill use, which was observed in the current study's sample.

Another study strength was measuring competency of facilitators. Because nonspecialist interventions are delivered by different types of personnel across settings, ranging from persons with only a high school education to college graduate nurses [47], it is important to establish minimum criteria on standardized competency measures across settings and facilitator types [48]. In this trial, we used ENACT, which is publicly available in a digital format through WHO Ensuring Quality in Psychological Support (EQUIP) platform [49]. By employing competency assessments, we excluded facilitators who had low skill levels after training and supplemented skill gaps for those who were retained. Through the competency assessments, we were able to observe that some nonspecialists displayed none or only a few of the foundational competencies (common factors for psychological interventions) prior to training. However, these individuals achieved competency on most skills by the end of training. Reporting competency levels achieved by facilitators in this trial allows future programs implementing Group PM+ to compare the competency of their facilitators and determine if their skills are adequate for safe and effective intervention delivery. If such monitoring is not done at scale, we cannot assume that the treatment effects that have been shown under research conditions will translate to real-world practice [48]. There is growing feasibility and expertise for this to be possible through the availability of a digital platform to guide implementation of competency-based

training and to provide instruction on training raters and actors to conduct competency evaluation [49]. Standardized role plays and structured competency rating for specific PM+ skills and group facilitation skills have also recently been developed to evaluate and assure minimum competency standards for effective services in individual and Group PM+ [50,51].

The study demonstrated change in a locally meaningful outcome: "heart–mind problems". Few prior studies have included locally salient outcomes [52], which are important to promote engagement, adherence, and scale-up, as well as minimize stigma [16,53]. The intervention and associated implementation materials also underwent a rigorous cultural adaptation process to assure validity of the concepts and strategies for Nepali language and culture [23]. The study also included detailed documentation of services received by the control condition and reasons for dropout throughout the study. Other strengths included procedures to minimize and monitor unblinding, high retention rates, and rarity of missing data.

Limitations include lack of power to evaluate gender-specific effects, as well as clustering of all male participants in one area of the district. Moreover, male participants on average were older. Therefore, we cannot make gender-specific conclusions about effectiveness. Future studies will need to adapt PM+ and test effectiveness for men. This is important because of the gap in interventions with demonstrated effectiveness specifically for men [3]. Regarding mediation, the tool used to evaluate mechanisms of action, the RTC, was developed and piloted during the preceding feasibility study [28]. However, the RTC has not been validated in other populations. For the statistical mediation analysis, we used the mean-centering mediation method that avoids the bias that can arise in the traditional non-centered approach. However, there are potential limitations to this approach. First, as with all mediation analyses, bias may arise if confounders of the mediator (midline RTC) and outcome (endline GHQ-12) relationship are missing from the model. Second, with smaller than anticipated overall effects of PM+ on endline GHQ-12, our estimation that 31% of that effect was mediated by midline RTC must be interpreted relative to the overall magnitude of effect. Relatedly, future studies should also consider more objective measures of mechanisms of action that are less subject to self-report bias. Future research should investigate long-term benefits (e.g., 12 months posttreatment), as well as the cost-effectiveness of Group PM+.

## Conclusions

A rigorously conducted cRCT evaluated the effectiveness of a brief group psychological intervention delivered by nonspecialists without prior mental health training and only a high school education level. We found modest benefits of Group PM+ compared to EUC. To increase the public health benefit of Group PM+, additional effort should be placed on strengthening PM+ skill use among participants. Future global mental health research should similarly attend to both competency and mechanisms of action to determine what works, how it works, and use this information to inform scaling up of psychological interventions in humanitarian emergencies and low-resource settings around the world.

## Supporting information

**S1 Text. RTC.** RTC, Reducing Tension Checklist.
(DOCX)

**S2 Text. CONSORT tables.** CONSORT, Consolidated Standards of Reporting Trials.
(DOCX)

**S3 Text. Supplementary analysis tables.**
(DOCX)

## Acknowledgments

We thank the research team of TPO Nepal for their support in the study. Special thanks to TPO Nepal's field supervision team: Ghanashyam Sharma, Ganesh Poudel, Bishnu Lamichhane, and Nutan Tiwari. ELT had full access to all the data in the study and takes responsibility for the integrity of the data and the accuracy of the data analysis.

## Disclaimers

The authors alone are responsible for the views expressed in this publication, and they do not necessarily represent the decisions, policy, or views of the World Health Organization.

## Author Contributions

**Conceptualization:** Mark J. D. Jordans, Brandon A. Kohrt, Elizabeth L. Turner, Edith van't Hof, Richard A. Bryant, Nagendra P. Luitel, Mark van Ommeren.

**Data curation:** Elizabeth L. Turner, Xueqi Wang.

**Formal analysis:** Elizabeth L. Turner, Xueqi Wang.

**Funding acquisition:** Mark van Ommeren.

**Investigation:** Mark J. D. Jordans, Brandon A. Kohrt, Manaswi Sangraula, Nagendra P. Luitel.

**Methodology:** Mark J. D. Jordans, Brandon A. Kohrt, Manaswi Sangraula, Elizabeth L. Turner, Pragya Shrestha, Edith van't Hof, Richard A. Bryant, Nagendra P. Luitel, Mark van Ommeren.

**Supervision:** Pragya Shrestha, Renasha Ghimire, Katie S. Dawson, Kedar Marahatta.

**Visualization:** Xueqi Wang.

**Writing – original draft:** Mark J. D. Jordans, Brandon A. Kohrt, Manaswi Sangraula, Elizabeth L. Turner.

**Writing – review & editing:** Xueqi Wang, Pragya Shrestha, Renasha Ghimire, Edith van't Hof, Richard A. Bryant, Katie S. Dawson, Kedar Marahatta, Nagendra P. Luitel, Mark van Ommeren.

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
