## [Editor Report · Decision Letter 0]

11 Nov 2020

Dear Dr Kohrt, 

Thank you for submitting your manuscript entitled "Effectiveness of a brief psychological intervention for adults affected by humanitarian disasters: a cluster randomized controlled trial of Group Problem Management Plus" for consideration by PLOS Medicine.

Your manuscript has now been evaluated by the PLOS Medicine editorial staff and I am writing to let you know that we would like to send your submission out for external assessment.

Kind regards,

Richard Turner, PhD

Senior editor, PLOS Medicine

rturner@plos.org

---

## [Decision Letter · Decision Letter 1]

16 Dec 2020

Dear Dr. Kohrt,

Thank you very much for submitting your manuscript "Effectiveness of a brief psychological intervention for adults affected by humanitarian disasters: a cluster randomized controlled trial of Group Problem Management Plus" (PMEDICINE-D-20-05408R1) for consideration at PLOS Medicine. 

Your paper was evaluated by the editors and sent to independent reviewers, including a statistical reviewer. The reviews are appended at the bottom of this email and any accompanying reviewer attachments can be seen via the link below:

[LINK]

In light of these reviews, we will not be able to accept the manuscript for publication in the journal in its current form, but we would like to invite you to submit a revised version that addresses the reviewers' and editors' comments fully. You will appreciate that we cannot make a decision about publication until we have seen the revised manuscript and your response, and we expect to seek re-review by one or more of the reviewers. 

We hope to receive your revised manuscript by Jan 13 2021 11:59PM. Please email us (plosmedicine@plos.org) if you have any questions or concerns.

Please let me know if you have any questions. Otherwise, we look forward to receiving your revised manuscript in due course. 

Sincerely,

Richard Turner, PhD

rturner@plos.org

Please remove the information on funding from the title page. In the event of publication, this information will appear in the article metadata via entries in the submission form. 

Please revisit your data statement. We would view publication under a CC BY licence as open-access publication (similar to the licence used at PLOS). You appear to be indicating that display items will be made available in this way, however: are you able to share patient-level data in spreadsheet(s)? Please note that article authors cannot serve as points of contact for inquiries about data access according to PLOS' data policy. 

In your title, we suggest moving "Group Problem Management Plus" before the colon. 

Please adapt the abstract to the three-part PLOS Medicine style. The final sentence of the "Methods and findings" subsection should begin "Study limitations include ..." or similar and quote 2-3 of the study's main limitations. 

After the abstract, we will need to ask you to add a new and accessible "Author summary" in non-identical prose. You may find it helpful to consult one or two recent research papers in PLOS Medicine to get a sense of the preferred style. 

In what form was informed consent obtained?

Please substitute "sex" for "gender" where appropriate. 

Please adapt reference call-outs throughout the manuscript to the following style: "... publicly available [4,17].", noting the absence of spaces within the square brackets. 

In your reference list, please convert all italics to plan text. Where appropriate, 6 author names should be listed rather than 3, followed by "et al.".

Please convert the attached CONSORT checklist to a stand-alone file, referred to in your methods section ("See S1_CONSORT_Checklist" or similar). In the checklist, please refer to individual items by section (e.g., "Results") and paragraph number rather than by page or line numbers, as the latter generally change in the event of publication. 

Comments from the reviewers:

*** Reviewer #1: 

[see attachment]

*** Reviewer #2: 

Statistical review

This paper reports a cluster randomised trial in Nepal investigating a brief psychological intervention for coping with emergency situations. The authors show a significant effect on the primary outcome, reproducing earlier trials. They also investigate the mechanism of action.

The statistical methods used are suitable and the trial is reported well. I have some minor comments on the reporting below.

1. Abstract - this would be easier to follow if structured.

2. Abstract - From just the abstract I found the timing of the endpoint hard to follow as it switches between 20 weeks past baseline and 3 months (presumably 20 weeks post baseline is 3 months post treatment). I would recommend keeping the phrasing consistent within the abstract.

3. For at least the primary outcome I would recommend providing a p-value in addition to the confidence interval. 

4. I'd recommend clarifying psychosocial skill is measured in participants within both arms; if possible a bit more detail on what it is would be useful.

5. Page 9: a couple of the secondary endpoints mentioned here are not mentioned in the abstract - I would recommend ensuring that all secondary endpoints reported in the paper are mentioned in the abstract.

6. Page 10: "Sub-group analyses" seems to refer to a analysis population rather than what typically would be understood by subgroup analyses. The statistical analysis plan refers to this as a completors population which I think is a better way of phrasing it. 

7. I would recommend mentioning the gender subgroup analysis in the statistical section of the paper as this was pre-specified.

8. Page 11 "Sensitivity analyses additionally adjusted for predictors of missing outcomes." - can these be added?

9. Page 11 "Number of PM+ skills used" - I found it a bit confusing that there are 5 skills, but the RTC checklist was 10 items - is the mediating variable actually some score from the checklist or a number between 0 and 5?

10. Page 12: "There was no evidence of an interaction between gender and the treatment effect; and there was no suggestion of a benefit of Group PM+ for men in gender-specific sub-analyses" - I would recommend phrasing the second part of the sentence differently; the effect of the intervention appears to be consistent across the gender subgroups, although not significant in the male subgroup (which would be quite an underpowered test given it's only 20% of the sample size).

11. Page 13: I'm not too knowledgeable about mediation analysis so this may not be possible - can an uncertainty interval on the 31% proportion be provided?

James Wason

*** Reviewer #3: 

Thank you for the opportunity to review this manuscript describing a cluster randomized controlled trial evaluating the effectiveness of a brief psychological intervention for adults affected by humanitarian disasters in Nepal. This is a rigorous and well-conducted trial, and an excellent contribution to the global mental health literature, especially for advancing use of task sharing psychological interventions in humanitarian settings. This study is timely, and adds further impressive evidence in support use of the PM+ intervention. A few general comments to consider below:

- Were the facilitators compensated for delivering PM+? How were the facilitators recruited?

- Can more details about the credentials of the supervisors be provided? Were these members of the research team?

- The assessment of competency of the facilitators using ENACT is an important strength in this study, as well as the fidelity to PM+. One concern is the logistical challenges with competency rating, especially across so many sites. Can the authors comment on this and whether this poses challenges to scaling up the PM+ model following this trial?

- These findings are important, and add to the other recent trials of the PM+ program. Is there any plans to assess longer term outcomes and sustained benefits of this program beyond the 3-month follow up?

- The benefits of PM+ are clear, though what are the expected challenges to implementation and sustained delivery of this program in routine settings? There is no mention of costs in this study. Were costs collected? If not, this should be highlighted as a limitation and important area to expand on this work for supporting adoption of this program within health systems globally.

***

[LINK]

---

## [Decision Letter · Decision Letter 2]

29 Mar 2021

Dear Dr. Kohrt,

Thank you very much for re-submitting your manuscript "Effectiveness of Group Problem Management Plus a brief psychological intervention for adults affected by humanitarian disasters in Nepal: a cluster randomized controlled trial" (PMEDICINE-D-20-05408R2) for consideration at PLOS Medicine. We do apologize for the delay in sending you a response. 

I have discussed the paper with editorial colleagues and our academic editor, and it was also seen again by one reviewer. I am pleased to tell you that, provided the remaining editorial and production issues are dealt with, we expect to be able to accept the paper for publication in the journal.

[LINK]

Please let me know if you have any questions in the meantime, and we look forward to receiving the revised manuscript shortly.   

Sincerely,

Richard Turner, PhD

rturner@plos.org

Requests from Editors:

Please finalize the information in your data statement (submission form). 

Please add a comma to your title after "Plus". 

Please move the Acknowledgements section from the title page to the end of the main text; and the information on Ethics to your Methods section. The information on data availability and trial registration can be removed, as this is available elsewhere in the article or metadata.

Please quote summary demographic information for participants in the "Methods and findings" subsection of your abstract.

After quoting the primary endpoint findings in the abstract, we suggest amending the text to "Among the secondary endpoints, Group PM+ was associated at endline with a larger proportion of ..." or similar. 

Please adapt the "Conclusions" subsection of your abstract to the style: "In this study, we found that ... modestly reduced ...", or similar.

Please trim the "What did the authors do and find" subsection of your author summary to 3-4 points, and we suggest removing some of the practical details of the study (which are also described in the abstract). 

Please update references 50 and 51.

Please rename Fig 1 "Participant flowchart ...".

Comments from Academic editor:

This is a significantly improved manuscript, and the authors were responsive to all of the reviewer comments. This is a rigorous and well-conducted cluster RCT of the PM+ intervention in humanitarian settings in Nepal. I only have one remaining comment in relation to the interpretation of the mediation outcome, which I believe may benefit from further statistical review.

Referring to Table 4, it looks like the scores on the RTC mediator measure barely change within and between study arms, and it also looks as though the baseline scores on the RTC are considerably higher in the PM+ group compared to the control group. My first question is whether these baseline differences between groups were factored into the mediation analyses? Also, with such small changes in this measure, is it possible to rule out that this was due to chance as opposed to the effects of the intervention? I believe that this highlights an important potential mechanism for the PM+ intervention; though, the authors may overstate the importance of this finding. Moreover, the RTC measure does not appear to have been previously validated, and was developed only for this study. Therefore, I believe more caution is warranted for interpreting the mediation analysis findings throughout this manuscript.

One minor point, in the Instruments section on page 12, the authors refer to the PHQ-9 as the "Primary Health Questionnaire" for measuring depression. This should be changed to "Patient Health Questionnaire".

Comments from Reviewers:

*** Reviewer #2: 

Thank you to the authors for addressing my previous comments well. I agree with their responses and have no further issues to raise.

***

[LINK]

---

## [Editor Report · Decision Letter 3]

12 Apr 2021

Dear Dr Kohrt, 

On behalf of my colleagues and the Academic Editor, Dr Naslund, I am pleased to inform you that we have agreed to publish your manuscript "Effectiveness of Group Problem Management Plus, a brief psychological intervention for adults affected by humanitarian disasters in Nepal: a cluster randomized controlled trial" (PMEDICINE-D-20-05408R3) in PLOS Medicine.

Prior to final acceptance, please: remove reference 51 if this is not accepted prior to final acceptance of your paper; and remove "the authors declare no conflict of interest" from the acknowledgements (this information will be conveyed in the article metadata).

PRESS

Sincerely, 

Richard Turner, PhD 

rturner@plos.org